# OpenReview forum: "Uncertainty and Influence aware Reward Model Refinement for Reinforcement Learning from Human Feedback"
_ICLR.cc/2025/Conference — ICLR 2025 Poster_

### Official Review · Reviewer_V4Mi · 2024-10-31

**Soundness:** 3
**Presentation:** 2
**Contribution:** 3
**Rating:** 8
**Confidence:** 4

**Summary:**

This paper presents a new method for improving reward model training by prioritising on-policy samples to relable to get the highest performance with the least relabelling cost. The method uses a combination of empirical uncertainty estimated by an ensemble of reward models, and gradient-based influence calculation to select data. The paper presents empirical results showing that their method outperforms baselines across a range of metrics in the HH dataset for two different reward model sizes. Their method improves both reward model performance and policy performance, and in a series of ablations the paper shows that all components of the proposed method are important for the success of the method.

**Strengths:**

The proposed method is interesting and novel, and shows strong improvements over reasonable baselines. The problem of reward model overoptimisation is an important and timely one, making the contribution reasonably significant. While the method adds additional complexity on top of the baselines, this does seem to pay off in improved performance, and ablations demonstrate that how each component contributes. The idea is intuitive which makes the contribution more compelling. The paper is mostly easy to read and understand. The contribution seems sufficiently original.

**Weaknesses:**

# lack of clarity on RLR

The RLR baseline is not explained in the main body of the paper. Could you add a description of this algorithm so that it's clear what UGDA is being compared to?

# Limited evaluation dataset

The experiments are only performed on anthropic HH. It would be beneficial to demonstrate improvements on another RLHF dataset as well. This is partially remedied by showing improvements on a range of metrics and for reward models as well as policies, but this is still a weakness of the paper.

# Lack of error bars

It would be useful to have error bars or confidence intervals on any of the reported results. I acknowledge that running experiments for additional seeds is costly, but it would make the results much more robust

# Summary

Overall I give this paper a 6. I would be willing to raise my score to a 7 if the clarity and error bars issues mentioned above were fixed, and I would raise my score higher if additional datasets where considered and results where still positive for UGDA.

# Update

Given the improved results and clarity fixes in the rebuttal, I am raising my score to an 8, assuming all of that is in the final version of the paper.

**Questions:**

The RLR baseline is not explained in the main body of the paper. Could you add a description of this algorithm so that it's clear what UGDA is being compared to?

---

> ### Author Response · Authors · 2024-11-19
> **Responses to Reviewer V4Mi (1/2)**
>
> **Dear Reviewer V4Mi,**
>
> Thanks for your careful and insightful comments, we provide our responses to your questions in the following.
>
> **W1: lack of clarity on RLR**
>
> **A1:** The RLR uses the filtered instructions in the data selection stage, but the responses are the corresponding pairwise responses in the preference dataset; we use the samples in the preference dataset to refine the reward model ensembles.
>
> **W2: Limited evaluation dataset**
>
> **A2:** To answer this question, we conduct addtional experiments based on the ultra feedback dataset [1] to train the policy, the results are presented in the following.
>
> | Method | Alpaca Eval (LC\_Winrate) | Alpaca Eval (Winrate) | Arena Hard | MTBench 101 |
> |--------|:-------------------------:|:---------------------:|:----------:|:-----------:|
> | PPO    |            11.8           |          8.0          |    11.7    |     4.7     |
> | LCB    |            13.0           |          8.6          |    10.7    |     5.2     |
> | UWO    |            12.6           |          9.1          |    12.8    |     5.5     |
> | RLR    |            13.3           |          8.8          |    12.4    |     5.3     |
> | UGDA   |            **14.8**           |          **10.9**         |    **13.5**    |     **6.1**     |
>
> **Reward Model Evaluation Results**
>
> - **Helpful results**
>
> | Reward Model | Training Methods | Average\_Reward | Var\_Reward |
> |--------------|:----------------:|:---------------:|:-----------:|
> | Gemma 2B     |        PPO       |       0.66      |     0.12    |
> |              |        LCB       |       0.61      |     0.13    |
> |              |        UWO       |       0.65      |     0.17    |
> |              |        RLR       |       0.74      |     0.15    |
> |              |       UGDA       |       0.91      |     0.09    |
> | Gemma 7B     |        PPO       |       0.73      |     0.14    |
> |              |        LCB       |       0.73      |     0.13    |
> |              |        UWO       |       0.83      |     0.15    |
> |              |        RLR       |       0.82      |     0.12    |
> |              |       UGDA       |       **0.94**      |    **0.11**    |
>
> - **Harmless results**
>
> | Reward Model | Training Methods | Average\_Reward | Var\_Reward |
> |--------------|:----------------:|:---------------:|:-----------:|
> | Gemma 2B     |        PPO       |       0.64      |     0.18    |
> |              |        LCB       |       0.63      |     0.14    |
> |              |        UWO       |       0.65      |     0.16    |
> |              |        RLR       |       0.70      |     0.17    |
> |              |       UGDA       |       **0.84**      |     **0.13**    |
> | Gemma 7B     |        PPO       |       0.74      |     0.14    |
> |              |        LCB       |       0.78      |     0.13    |
> |              |        UWO       |       0.72      |     0.13    |
> |              |        RLR       |       0.80      |      **0.11**     |
> |              |       UGDA       |       **0.93**      |    0.12   |
>
> **GPT Comparision Results**
>
> |              |  Win  |  Tie  |  Lose |
> |--------------|:-----:|:-----:|:-----:|
> | UGDA vs. PPO | 49.6% | 24.5% | 25.9% |
> | UGDA vs. LCB | 51.2% | 16.6% | 32.2% |
> | UGDA vs. UWO | 46.7% | 21.8% | 31.5% |
> | UGDA vs. RLR | 49.8% | 19.8% | 30.4% |
>
> From the above results, we can see that UGDA still outperforms than all the baselines, which further verify the effectiveness of our architecture. Additionally, we do not apply different random seeds for the new dataset experiments due to the time limitation of the rebuttal. If needed, we will add the results with the error bars.
>
> **References**
>
> [1] Cui, G., Yuan, L., Ding, N., Yao, G., He, B., Zhu, W., ... & Sun, M. (2024, June). ULTRAFEEDBACK: Boosting Language Models with Scaled AI Feedback. In Forty-first International Conference on Machine Learning.

---

> > ### Author Response · Authors · 2024-11-19
> > **Responses to Reviewer V4Mi (2/2)**
> >
> > **W3: Lack of error bars**
> >
> > **A3:** We conduct additional experiments with different random seeds, specially, we use five different random seeds for the policy optimization. The results are shown in the following.
> >
> > **Benchmark Evaluation Results**
> >
> > | Method | Alpaca Eval (LC\_Winrate) | Alpaca Eval (Winrate) |    Arena Hard    |   MTBench 101   |
> > |--------|:-------------------------:|:---------------------:|:----------------:|:---------------:|
> > | PPO    |        11.7 ± 0.015       |      8.0 ± 0.010      |   11.7 ± 0.021   |   4.7 ± 0.018   |
> > | LCB    |        12.8 ± 0.009       |      8.6 ± 0.014      |   10.7 ± 0.014   |   5.2 ± 0.013   |
> > | UWO    |        12.4 ± 0.012       |      9.1 ± 0.013      |   12.8 ± 0.015   |   5.5 ± 0.008   |
> > | RLR    |        13.1 ± 0.020       |      8.8 ± 0.016      |   11.7 ± 0.009   |   5.3 ± 0.017   |
> > | UGDA   |      **14.2 ± 0.011**     |    **10.8 ± 0.018**   | **13.4 ± 0.011** | **6.2 ± 0.012** |
> >
> > **Reward Model Evaluation Results**
> >
> > - **Helpful results**
> >
> > | Reward Model | Training Methods |  Average\_Reward  |    Var\_Reward    |
> > |--------------|:----------------:|:-----------------:|:-----------------:|
> > | Gemma 2B     |        PPO       |   0.61 ± 0.0372   |   0.19 ± 0.0231   |
> > |              |        LCB       |   0.52 ± 0.0401   |   0.19 ± 0.0196   |
> > |              |        UWO       |   0.53 ± 0.0383   |   0.20 ± 0.0228   |
> > |              |        RLR       |   0.71 ± 0.0219   |   0.17 ± 0.0390   |
> > |              |       UGDA       | **0.80 ± 0.0232** | **0.12 ± 0.0289** |
> > | Gemma 7B     |        PPO       |   0.62 ± 0.0398   |   0.22 ± 0.0121   |
> > |              |        LCB       |   0.74 ± 0.0384   |   0.18 ± 0.0212   |
> > |              |        UWO       |   0.70 ± 0.0403   |   0.16 ± 0.0426   |
> > |              |        RLR       |   0.80 ± 0.0354   |   0.15 ± 0.0399   |
> > |              |       UGDA       | **0.89 ± 0.0492** | **0.13 ± 0.0276** |
> >
> > - **Harmless results**
> >
> > | Reward Model | Training Methods |  Average\_Reward  |    Var\_Reward    |
> > |--------------|:----------------:|:-----------------:|:-----------------:|
> > | Gemma 2B     |        PPO       |   0.63 ± 0.0239   |   0.19 ± 0.0393   |
> > |              |        LCB       |   0.58 ± 0.0377   |   0.22 ± 0.0411   |
> > |              |        UWO       |   0.60 ± 0.0288   |   0.17 ± 0.0407   |
> > |              |        RLR       |   0.62 ± 0.0297   |   0.14 ± 0.0342   |
> > |              |       UGDA       | **0.79 ± 0.0197** | **0.13 ± 0.0296** |
> > | Gemma 7B     |        PPO       |   0.72 ± 0.0377   |   0.21 ± 0.0223   |
> > |              |        LCB       |   0.74 ± 0.0343   |   0.17 ± 0.0356   |
> > |              |        UWO       |   0.71 ± 0.0268   |   0.13 ± 0.0402   |
> > |              |        RLR       |   0.72 ± 0.0372   |   0.15 ± 0.0389   |
> > |              |       UGDA       | **0.86 ± 0.0474** | **0.11 ± 0.0322** |
> >
> > **GPT Comparision Results**
> >
> > |              |      Win     |      Tie     |     Lose     |
> > |--------------|:------------:|:------------:|:------------:|
> > | UGDA vs. PPO | 51.3% ± 6.62 | 21.2% ± 4.47 | 27.5% ± 5.67 |
> > | UGDA vs. LCB | 49.5% ± 3.36 | 13.4% ± 3.83 | 37.1% ± 7.39 |
> > | UGDA vs. UWO | 48.6% ± 5.07 | 20.3% ± 2.15 | 31.1% ± 5.51 |
> > | UGDA vs. RLR | 53.9% ± 4.07 | 21.1% ± 2.74 | 25.0% ± 4.45 |
> >
> > From the above results, we can know that the superiority of our UGDA is not affected by random seed, and we will add the above results to our final version.

---

> > > ### Comment · Reviewer_V4Mi · 2024-11-21
> > >
> > > Thanks for your response. Given the improved results and clarity fixes in the rebuttal, I am raising my score to an 8, assuming all of that is in the final version of the paper.

---

> > > > ### Author Response · Authors · 2024-11-21
> > > >
> > > > Thanks for your positive feedback! We will follow your advice and add all the content in our final version.

---

### Official Review · Reviewer_Jzfj · 2024-11-03

**Soundness:** 3
**Presentation:** 2
**Contribution:** 2
**Rating:** 6
**Confidence:** 3

**Summary:**

This paper presents Uncertainty-Gradient-based Data Augmentation (UGDA), an approach to enhance reinforcement learning from human feedback for training LLMs. UGDA addresses a key issue: the limitations of an imperfect reward model that serves as a proxy for human preferences, which becomes increasingly problematic as policy optimization alters the training data distribution. UGDA aims to refine the reward model by selectively relabeling the most uncertain and impactful interaction data during policy optimization. The proposed approach follows a three-stage pipeline: Reward LoRA Ensembles, Data Selection, and Reward LoRA Refinement. Experiments demonstrate that UGDA improves policy performance without the costly requirement of additional human preference data collection.

**Strengths:**

1. The paper introduces a new method for selecting the uncertain and impactful interaction data during policy optimization to refine the reward model.
2. The paper is well-organised and articulated with clarity, the complex concepts are explained in a clear and concise manner.
3. The results, partly supported by some confidence/uncertainty based methods, demonstrate considerable potential for improving alignment of LLMs with human values.

**Weaknesses:**

The results have a few issues which make evaluating the contribution difficult:
1. The paper lacks a comparison with some existing works, particularly methods involve iterative PPO/DPO method that train a reward model simultaneously and  reward ensembles [1].
[1] Coste T, Anwar U, Kirk R, et al. Reward model ensembles help mitigate overoptimization.
2. The alignment of relabeled reward data with human annotator judgments remains insufficiently validated.

**Questions:**

Could the authors provide a more detailed description of the process for selecting the validation dataset? Given that the impact score depends on the validation set, what are the implications if the validation dataset distribution significantly differs from the training set? Will this also introduce high bias in the impact score.

How does the computational complexity of UGDA scale with model and dataset size, considering the use of an ensemble of reward models and the validation dataset?

What if the variance in reward signals provided by different reward models for the same questions and responses remains high after refining the reward models? The paper lacks an analysis of how refining the reward models impacts the consistency of the reward signals.

---

> ### Author Response · Authors · 2024-11-19
> **Responses to Reviewer Jzfj (1/2)**
>
> **Dear Reviewer Jzfj,**
>
> Thanks for your careful and insightful comments, we provide our responses to your questions in the following.
>
> **W1: Comparison with iterative PPO/DPO method and reward ensembles.**
>
> **A1:** We have conducted the reward ensembles method UWO [1] in the current version of our paper as the baseline.
> Additionally, follow your advice, we conduct the experiment based on the DPO [2] and iterative DPO [3], the results are based on the Gemma 7B policy.
>
> **Benchmark Evaluation Results**
>
> | Method        | Alpaca Eval (LC\_Winrate) | Alpaca Eval (Winrate) | Arena Hard | MTBench 101 |
> |---------------|:-------------------------:|:---------------------:|:----------:|:-----------:|
> | DPO           |            11.1           |          7.9          |    10.8    |     4.9     |
> | iterative DPO |            11.9           |          8.3          |    11.7    |     5.2     |
> | UGDA          |            **14.2**           |          **10.7**         |    **13.6**    |     **6.1**     |
>
> **Reward Model Evaluation Results**
> | Test Setting  | Training Methods | Average\_Reward | Var\_Reward |
> |---------------|:----------------:|:---------------:|:-----------:|
> | Helpful    |        DPO       |       0.61      |     0.15    |
> |      |   iterative DPO  |       0.69      |     0.14    |
> |       |       UGDA       |       **0.91**      |     **0.12**    |
> | Harmless      |        DPO       |       0.58      |     0.13    |
> |               |   iterative DPO  |       0.65      |     0.12    |
> |               |       UGDA       |       **0.88**     |     **0.10**    |
>
> **GPT Comparision Results**
>
> |                        |  Win  |  Tie  |  Lose |
> |------------------------|:-----:|:-----:|:-----:|
> | UGDA vs. DPO           | 51.8% | 23.9% | 24.3% |
> | UGDA vs. Iterative DPO | 48.9% | 21.7% | 29.4% |
>
> From the results, we can see that iterative DPO outperforms DPO, but is less effective than UGDA. This may be because of the superior generalization ability of PPO compared with DPO.
>
> **References**
>
> [1] Coste, T., Anwar, U., Kirk, R., & Krueger, D. (2023). Reward model ensembles help mitigate overoptimization. arXiv preprint arXiv:2310.02743.
>
> [2] Rafailov, R., Sharma, A., Mitchell, E., Manning, C. D., Ermon, S., & Finn, C. (2024). Direct preference optimization: Your language model is secretly a reward model. Advances in Neural Information Processing Systems, 36.
>
> [3] Xu, J., Lee, A., Sukhbaatar, S., & Weston, J. (2023). Some things are more cringe than others: Preference optimization with the pairwise cringe loss. arXiv preprint arXiv:2312.16682.
>
> **W2: Relabeled reward data validation.**
>
> **A2:** We provide the comparision of the relabeled reward data with human or GPT annotator in Table 1, which shows the similarity of the human labelers and GPT-4, which is suffecient for supporting us to use GPT annotator in the following experiments. If needed, we are willing to conduct more experiments to verify the results with your advice.
>
> **Q1: Validation dataset construction.**
>
> **A1:** We build the validation dataset by randomly selecting 10% samples of the perference dataset, specially each samople is constructed by the instruction with chosen response. We will add the related description in the experimental setup in our final version.
>
> **Q2: The computational complexity.**
>
> A2: The complexity of **reward model training with ensembles** scales with the preference data size $\mathcal{D}_p$ and the number of ensembles $K$, which is $\mathcal{O}(|\mathcal{D}_p|\cdot K)$.
>
> For the gradient based data selection, we divide this into two stage to analysis the complexity. The **gradient feature computation** is the most expensive step, and the cost scales linearly with candidate dataset size $|\mathcal{D}\_{\text{train}}|$, the number of checkpoints $N$ and the projection dimension $d$, which is $\mathcal{O}(|\mathcal{D}\_{\text{train}}|\cdot N \cdot d)$. From the experimental results in our work,  we can see that a small projection dimension is sufficient.
>
> The **data selection** process requires minimal computation, and scales with the size of $|\mathcal{D}\_{\text {train}}|$ and $|\mathcal{D}\_{\text{val}}|$, which  is $\mathcal{O}\left(|\mathcal{D}\_{\text {train}}| \cdot\left|\mathcal{D}\_{\text {val }}\right|\right)$.
>
> Furthermore, we will add the above analysis to our final version.

---

> > ### Author Response · Authors · 2024-11-19
> > **Responses to Reviewer Jzfj (2/2)**
> >
> > **Q3: Variance problem.**
> >
> > **A3:** To answer this question, we count the variance of all training samples. Before the iteration of reward model refining, the **smallest variance among the top 50% of samples is 0.24**. After refining, only **21% of the samples have a variance greater than 0.24**, indicating that the consistency of sample scoring by reward model ensembles has increased.
> >
> > The above results also mean that, if we want to get more consistent reward model ensembles, we can further introduce more iterations for the policy optimization and reward refining. But considering the time cost and the training difficulty of PPO, combining with the results in our work, we can observe that one iteration is enough for the better results to show the effectiveness of UGDA.

---

> > > ### Author Response · Authors · 2024-11-24
> > >
> > > **Dear Reviewer Jzfj,**
> > >
> > > In the above responses, we have try our best to answer your questions and solve your confusions. Due to the rebuttal ddl is coming, we are very willing to have a more in-depth discussion with you, and we welcome you to give us more suggestions. If you have additional suggestions, please let us know and we will try to respond as quickly as possible.

---

> > > > ### Comment · Reviewer_Jzfj · 2024-11-25
> > > > **Official comment by reviewer**
> > > >
> > > > The reply did not address all of my concerns, particularly, I still believe that selectively relabeling the most uncertain and impactful interaction data during policy optimization could be computationally costly. However, based on the detailed experiments provided, I have decided to raise my score.

---

> > > > > ### Author Response · Authors · 2024-11-25
> > > > >
> > > > > Thanks for your positive feedback, we will add more details in our final version, and we will further improve our method in future work.

---

### Official Review · Reviewer_DX3G · 2024-11-03

**Soundness:** 3
**Presentation:** 3
**Contribution:** 2
**Rating:** 6
**Confidence:** 2

**Summary:**

This paper study the reward modeling using LLMs in RLHF. The authors argue that the existing reward models, which serve as proxies for human preferences, become less effective as the policy optimization process shifts the distribution of the training dataset. To address this, UGDA leverages policy interaction samples, selecting them based on the uncertainty of the reward ensembles and the gradient-based influence on policy optimization. The selected samples are then used to refine the reward model through supervised learning, which in turn improves the policy. Empirical results show that UGDA can enhance policy performance without the need for costly human preference data collection, surpassing state-of-the-art methods in experiments.

**Strengths:**

This paper is well-motivated and well-written.

This paper aims to address the off-distribution reward overoptmization problem, which is a critical issue in RLHF.

**Weaknesses:**

Lacks comparisons to active learning and data augmentation baselines.

Considering that off-policy methods have become popular, additional analysis of the robustness of UGDA under different policy optimization scenarios, such as iterative DPO, could strengthen the paper.

**Questions:**

Could you apply the method to the full fine-tuning reward ensembles?

---

> ### Author Response · Authors · 2024-11-19
> **Responses to Reviewer DX3G**
>
> **Dear Reviewer DX3G,**
>
> Thanks for your careful and insightful comments, we provide our responses to your questions in the following.
>
> **W1: Lacks comparisons to active learning and data augmentation baselines.**
>
> **A1:** We conduct experiments with the ablation study which only uses uncertainty. This can be seen as an active learning-based method.
>
> To achieve the fair comparision, we have conducted the experiments based on the random data augmentation as the baseline of our method in the experiments, which can be seen as the data augmentation baseline. The results are shown in Section 5.3.
>
> **W2: Off-policy methods.**
>
> **A2:** We follow your advice to conduct experimental analysis based on the DPO [1] and iterative DPO [2]. Specially, the for the iterative DPO, to achieve the fair comparison, we only apply **one round iteration with 25% samples** for data augmentation, we use GPT annotator as the reward model for iterative DPO. Moreover, the experimental results are based on the **Gemma 7B policy**.
>
> **Benchmark Evaluation Results**
>
> | Method        | Alpaca Eval (LC\_Winrate) | Alpaca Eval (Winrate) | Arena Hard | MTBench 101 |
> |---------------|:-------------------------:|:---------------------:|:----------:|:-----------:|
> | DPO           |            11.1           |          7.9          |    10.8    |     4.9     |
> | iterative DPO |            11.9           |          8.3          |    11.7    |     5.2     |
> | UGDA          |            **14.2**           |          **10.7**         |    **13.6**    |     **6.1**     |
>
> **Reward Model Evaluation Results**
> | Test Setting  | Training Methods | Average\_Reward | Var\_Reward |
> |---------------|:----------------:|:---------------:|:-----------:|
> | Helpful       |        DPO       |       0.61      |     0.15    |
> |               |   iterative DPO  |       0.69      |     0.14    |
> |               |       UGDA       |        **0.91**      |      **0.12**    |
> | Harmless      |        DPO       |       0.58      |     0.13    |
> |               |   iterative DPO  |       0.65      |     0.12    |
> |               |       UGDA       |       **0.88**     |     **0.10**    |
>
> **GPT Comparision Results**
>
> |                        |  Win  |  Tie  |  Lose |
> |------------------------|:-----:|:-----:|:-----:|
> | UGDA vs. DPO           | 51.8% | 23.9% | 24.3% |
> | UGDA vs. Iterative DPO | 48.9% | 21.7% | 29.4% |
>
> From the results, we can see that iterative DPO outperforms DPO, but is less effective than UGDA. This may be because of the superior generalization ability of PPO compared with DPO.
>
> **References**
>
> [1] Rafailov, R., Sharma, A., Mitchell, E., Manning, C. D., Ermon, S., & Finn, C. (2024). Direct preference optimization: Your language model is secretly a reward model. Advances in Neural Information Processing Systems, 36.
>
> [2] Xu, J., Lee, A., Sukhbaatar, S., & Weston, J. (2023). Some things are more cringe than others: Preference optimization with the pairwise cringe loss. arXiv preprint arXiv:2312.16682.
>
> **Q1: Could you apply the method to the full fine-tuning reward ensembles?**
>
> **A1:**  Our method can be directly applied to the full fine-tuning without additional modification, but due to the computational cost and the time limitation, we can not provide the fine-tuning results during the rebuttal,  we are sorry for this. From the results of the LoRA fine-tuning, the effectiveness of our UGDA can be verified.

---

> > ### Author Response · Authors · 2024-11-22
> > **Additional Responses to Q1 (full fine-tuning)**
> >
> > **Dear Reviewer DX3G,**
> >
> > In addition to the above response, we re-evaluated the feasibility of the full fine-tuning experiments, and try our best to conduct the experiments to answer Q1.
> >
> > But due to some limitation about the computational and time cost, we use another small base model to implement the baselines and our method.
> >
> > Specifically, in the full fine-tuning process, we use the **Pythia 70M model as the reward model and the Pythia 1.4B model as the policy model**. The **reward judge model is the Pythia 6.9B**. We regret that, due to limited time constraints, we are unable to train a larger model for full fine-tuning.
> >
> > **Benchmark Results**
> > | Method | Alpaca Eval (LC\_Winrate) |  Alpaca Eval (Winrate) | Arena Hard | MTBench 101 |
> > |---|:---: |:---: |:---:|:---:|
> > | PPO | 0.31  |0.17  |0.34  |0.16 |
> > | LCB | 0.33 |0.19 |0.41|0.19  |
> > | UWO | 0.42 |0.22 |0.37| 0.17  |
> > | RLR | 0.44|0.24|0.43|0.22 |
> > | UGDA | **0.65** |**0.32** | **0.52** | **0.26** |
> >
> > **Reward Model Evaluation Results**
> >
> > | Test Setting|Training Methods|Average\_Reward|Var\_Reward|
> > |---|:---: |:---: |:---:|
> > |Helpful | PPO | 0.48 | 0.22|
> > | |LCB | 0.45 |0.24|
> > | |UWO | 0.53 |0.21|
> > | |RLR | 0.59 |0.19|
> > | |UGDA| **0.66** |**0.16**|
> > |Harmless | PPO | 0.51 | 0.26|
> > | |LCB | 0.49 |0.25|
> > | |UWO | 0.55 |0.24|
> > | |RLR | 0.64 |0.26|
> > | |UGDA| **0.68** |**0.19**|
> >
> >
> >
> >
> > **GPT comparision Results**
> >
> > || Win |Tie |Lose|
> > |---|:---: |:---: |:---:|
> > |UGDA vs. PPO| 38.5%  | 35.3% | 26.2% |
> > |UGDA vs. LCB | 41.7%  | 36.6%  | 21.7% |
> > |UGDA vs. UWO | 39.6% | 31.2%  | 29.2% |
> > |UGDA vs. RLR | 34.7%  | 33.5%  | 31.8% |
> >
> > From the results, it is evident that our UGDA outperforms various baselines. However, compared to the results on the Gemma model, the improvement is less significant. This could be attributed to **the small size of the base model and the capabilities of the Pythia model**. Nevertheless, the experimental results still **support the superiority of our UGDA in the full fine-tuning scenario**.
> >
> >
> > Moreover, we are very willing to have a more in-depth discussion with you, and we welcome you to give us more suggestions. If you have additional suggestions, please let us know and we will try to respond as quickly as possible.

---

> > > ### Author Response · Authors · 2024-11-24
> > >
> > > **Dear Reviewer DX3G,**
> > >
> > > In the above responses, we have try our best to answer your questions and solve your confusions. Due to the rebuttal ddl is coming, we are very willing to have a more in-depth discussion with you, and we welcome you to give us more suggestions. If you have additional suggestions, please let us know and we will try to respond as quickly as possible.

---

> > > > ### Comment · Reviewer_DX3G · 2024-11-26
> > > >
> > > > Thanks for your response. My concerns are addressed and I have raised my score to 6.

---

> > > > > ### Author Response · Authors · 2024-11-26
> > > > >
> > > > > Thanks for your positive feedback! We will follow your advice and add more details in our final version.

---

### Official Review · Reviewer_wH1u · 2024-11-04

**Soundness:** 3
**Presentation:** 2
**Contribution:** 3
**Rating:** 8
**Confidence:** 4

**Summary:**

The paper considers the reward learning problem in reinforcement learning from human feedback (RLHF). As a reward model is trained from preference data and can be inaccurate, it would be beneficial to allow the model to query the labels of some samples. The key question is: querying the labels of which samples can improve the model’s performance the most.

The paper proposes Uncertainty-Gradient based Data Augmentation (UGDA). The method selects samples for labeling based on a joint objective combining (1) the uncertainty of the sample’s prediction (based on an ensemble of reward models) and (2) the influence of the sample on a validation set. The model identifies samples that maximize this joint objective and queries their labels. Empirical results demonstrate that UGDA outperforms other sample selection methods in improving model performance.

**Strengths:**

Novelty: The algorithm is novel in using gradient-based data influence for reward learning. To the best of my knowledge, this approach has not been previously applied to reward learning in RLHF.

Motivation and Empirical Performance: The proposed method is well-motivated and well explanable. Empirical results show UGDA's effectiveness.

**Weaknesses:**

**Motivation and algorithm design.** I wonder if UGDA can be over-complicated for the goal it wants to achieve. For example, can I use simpler heuristics to select samples that may lead to a larger learning gradients?

For example, does UGDA tend to select out-of-distribution samples in the training set (as these samples can lead to larger training gradients)? If we simply find samples that are outliers in the training dataset (for example, using data density distribution), would they perform similarly to UGDA?

**Evaluation.** The reward models are all finetuned from Gemma models, and Llama-2-13B model is used for evaluation. Would UGDA have similar results if tested with different model architectures?

**Presentation.**
Notational Clarity: The notation could be improved for clarity. For example, $R_v$ looks like a specific reward function, but it actually represents variance.

The paper would benefit from overall improvements in clarity and grammar. Specific issues include:
Line 18-19, ungrammatical: “Since the policy optimization continuously shifts the human preference training dataset’s distribution.”
Line 48 “Recently, there are some works [that] focus on”

**Questions:**

I have some questions in the weakness section above.

I didn’t see details on how the validation set is constructed. Does the validation set have the same distribution as the training set?

---

> ### Author Response · Authors · 2024-11-19
> **Responses to Reviewer wH1u (1/2)**
>
> **Dear Reviewer wH1u,**
>
> Thanks for your careful and insightful comments, we provide our responses to your questions in the following.
>
> **W1: Motivation and algorithm design**
>
> **A1:** In the training procedure of RLHF, for the gradient-based data selection, our motivation is to identify the most relevant data from these extensive datasets to effectively develop specific capabilities, which can be used for the further reward model refining. Specifically, we study the problem of identifying the influence of training samples on the prediction of the validation samples. The data influence can be learned by gradient information [1,2].
>
> Additionally, our experiments based on the complex QA tasks, thus we can not directly define the samples that are out of distribution by the data visualization. Thus, we use a heuristic way to define the out of distribution samples by **the length of the instruction**, specially, to achieve the fair comparision, we sort the samples by the length of the instructions, we use **25% of them to be relabeled by the GPT annotator** and further refine the reward models. We show the results of the policy optimization in the following, and we use OOD (length) to represent the above method.
> | Method | Alpaca Eval (LC\_Winrate) |  Alpaca Eval (Winrate) | Arena Hard | MTBench 101 |
> |---|:---: |:---: |:---:|:---:|
> | OOD (length) | 12.4 |8.3 |11.9 |5.2 |
> | UGDA | **14.2**  |**10.7**  | **13.6**  | **6.1**  |
>
> From the results, we can observe that, if we use the length of instructions as the criterion to select the samples, the results are not much more superior than the random selection for the policy optimizaiton. This may be because the samples that influence the gradient may not only be the OOD samples, which means, in the policy optimization process, the  learned policy can generalize to some OOD scenarios. Thus, these OOD samples may not be the most important factor affecting the gradient.
>
> **References**
>
> [1] Pruthi, G., Liu, F., Kale, S., & Sundararajan, M. (2020). Estimating training data influence by tracing gradient descent. Advances in Neural Information Processing Systems, 33, 19920-19930.
>
> [2] Han, X., Simig, D., Mihaylov, T., Tsvetkov, Y., Celikyilmaz, A., & Wang, T. (2023). Understanding in-context learning via supportive pretraining data. arXiv preprint arXiv:2306.15091.
>
> **W2: Evaluation**
>
> **A2:** As shown in the scaling law of the reward model [1], the bigger reward model has strong ability, thus we use it to evaluate the result. To answer this question, we additionally use the Gemma-2 27B to evaluate the results.
>
> - **Helpful results**
>
> | Reward Model | Training Methods | Average\_Reward | Var\_Reward |
> |--------------|:----------------:|:---------------:|:-----------:|
> | Gemma 2B     |        PPO       |       0.53      |     0.17    |
> |              |        LCB       |       0.57      |     0.15    |
> |              |        UWO       |       0.44      |     0.14    |
> |              |        RLR       |       0.48      |     0.09    |
> |              |       UGDA       |       **0.69**     |     **0.08**    |
> | Gemma 7B     |        PPO       |       0.55      |     0.19    |
> |              |        LCB       |       0.63      |     0.16    |
> |              |        UWO       |       0.64      |     0.17    |
> |              |        RLR       |       0.73      |     0.13    |
> |              |       UGDA       |       **0.84**      |     **0.10**   |
>
> - **Harmless results**
>
> | Reward Model | Training Methods | Average\_Reward | Var\_Reward |
> |--------------|:----------------:|:---------------:|:-----------:|
> | Gemma 2B     |        PPO       |       0.51      |     0.13    |
> |              |        LCB       |       0.48      |     0.15    |
> |              |        UWO       |       0.49      |     0.14    |
> |              |        RLR       |       0.55      |     0.12    |
> |              |       UGDA       |       **0.70**      |    **0.09**    |
> | Gemma 7B     |        PPO       |       0.67      |     0.14    |
> |              |        LCB       |       0.71      |     0.12    |
> |              |        UWO       |       0.68      |     0.11    |
> |              |        RLR       |       0.70      |     0.11    |
> |              |       UGDA       |       **0.81**      |     **0.07**    |
>
> From the results, we can see that our UGDA still outperform than other baselines, the results show that our method can acheive better performance with different bigger reward model evaluation, which is independent of the specific model structure.
>
> **References**
>
> [1] Gao, L., Schulman, J., & Hilton, J. (2023, July). Scaling laws for reward model overoptimization. In International Conference on Machine Learning (pp. 10835-10866). PMLR.
>
>
> **W3: Presentation.**
>
> **A3:** We will follow your advice and revise all the misleading presentations. And we will also check the full paper again to polish the presentation in our final version.

---

> > ### Author Response · Authors · 2024-11-19
> > **Responses to Reviewer wH1u (2/2)**
> >
> > **Q1: I didn’t see details on how the validation set is constructed. Does the validation set have the same distribution as the training set?**
> >
> > **A1:**  We build the validation dataset by randomly selecting 10% samples of the perference dataset $\mathcal{D}_p$, specially each sample is constructed by the instruction with chosen response. The validation set and training set have different distributions. The training dataset are the interaction data during policy optimization, but the validation dataset is formed by the preference dataset. Furthermore, we will add the related description to the experimental setup in our final version.

---

> > > ### Author Response · Authors · 2024-11-24
> > >
> > > **Dear Reviewer wH1u,**
> > >
> > > In the above responses, we have try our best to answer your questions and solve your confusions. Due to the rebuttal ddl is coming, we are very willing to have a more in-depth discussion with you, and we welcome you to give us more suggestions. If you have additional suggestions, please let us know and we will try to respond as quickly as possible.

---

> ### Comment · Reviewer_wH1u · 2024-11-25
>
> Thanks for providing additional experiment results and clarifications. The authors have addressed my concerns. The proposed method has more solid empirical justifications and I have raised the score.
>
> The paper can still benefit further from a clearer presentation in the revision.

---

> > ### Author Response · Authors · 2024-11-26
> >
> > Thanks for your positive feedback! We will follow your advice to add more details and further improve the presentation in our final version.

---

### Meta-Review · Area_Chair_fRZ4 · 2024-12-21

**Metareview:**

This paper studies reward modeling in RLHF. It uses uncertainty and gradient information to select important data, has further improved the learned reward model, and shows SoTA performance empirically.

Strengths:
This paper is well-written, novel, and well-motivated. The comprehensive experiments show strong performance.

Weaknesses:
Some reviewers questioned the motivation and algorithm design, and worried that the proposed method is over-complicated. Some other reviewers were concerned about the lack of sufficient experiments. However, most of these weaknesses were addressed during the rebuttal period.

All reviewers of this paper agreed that this paper made significant contributions to reward modeling in RLHF, so I would vote to accept.

**Additional Comments On Reviewer Discussion:**

During the rebuttal period, the authors provided extensive additional experimental results, and addressed most concerns from all reviewers.

---

### Decision · Program_Chairs · 2025-01-22

Accept (Poster)